# Immunopeptidome Diversity in Chronic Lymphocytic Leukemia Identifies Patients with Favorable Disease Outcome

**DOI:** 10.3390/cancers14194659

**Published:** 2022-09-25

**Authors:** Maddalena Marconato, Yacine Maringer, Juliane S. Walz, Annika Nelde, Jonas S. Heitmann

**Affiliations:** 1Clinical Collaboration Unit Translational Immunology, German Cancer Consortium (DKTK), Department of Internal Medicine, University Hospital Tübingen, 72076 Tübingen, Germany; 2Cluster of Excellence iFIT (EXC2180) “Image-Guided and Functionally Instructed Tumor Therapies”, University of Tübingen, Auf der Morgenstelle 15, 72076 Tübingen, Germany; 3Department of Peptide-Based Immunotherapy, University and University Hospital Tübingen, 72076 Tübingen, Germany; 4Institute for Cell Biology, Department of Immunology, University of Tübingen, 72076 Tübingen, Germany

**Keywords:** chronic lymphocytic leukemia, immunopeptidome, clinical outcome, survival

## Abstract

**Simple Summary:**

Immunosurveillance of cancer is mediated by T cell-based recognition of tumor-associated antigens, i.e., short peptides that are presented on the surface of cells on human leukocyte antigen (HLA) molecules. This encourages the analysis of the entirety of HLA-presented peptides, the so-called immunopeptidome, of malignant and benign cells, in order to identify novel therapeutic targets presented exclusively on malignant cells. In the present study, we aim to investigate the role of previously described immunopeptidome-defined antigen presentation in chronic lymphocytic leukemia (CLL) patients for clinical characteristics and disease outcome. We observed that higher yields of presented total and CLL-exclusive peptides were associated with a more favorable disease course, suggesting efficient immunosurveillance in a subgroup of patients and the possibility of further investigating T cell-based therapeutic approaches for CLL.

**Abstract:**

Chronic lymphocytic leukemia (CLL) is characterized by recurrent relapses and resistance to treatment, even with novel therapeutic approaches. Despite being considered as a disease with low mutational burden and thus poor immunogenic, CLL seems to retain the ability of eliciting specific T cell activation. Accordingly, we recently found non-mutated tumor-associated antigens to play a central role in CLL immunosurveillance. Here, we investigated the association of total and CLL-exclusive HLA class I and HLA class II peptide presentation in the mass spectrometry-defined immunopeptidome of leukemic cells with clinical features and disease outcome of 57 CLL patients. Patients whose CLL cells present a more diverse immunopeptidome experienced fewer relapses. During the follow-up phase of up to 10 years, patients with an HLA class I-restricted presentation of high numbers of total and CLL-exclusive peptides on their malignant cells showed a more favorable disease course with a prolonged progression-free survival (PFS). Overall, our results suggest the existence of an efficient T cell-based immunosurveillance mediated by CLL-associated tumor antigens, supporting ongoing efforts in developing T cell-based immunotherapeutic strategies for CLL.

## 1. Introduction

Chronic lymphocytic leukemia (CLL), the most common leukemia in adults in western countries, is characterized by the clonal expansion of a population of atypical B cells, which accumulates in the blood, bone marrow, lymph nodes and spleen [1,2,3]. CLL is a heterogeneous tumor entity, with survival ranging from approximately 2 to 20 years and occasionally presenting an aggressive disease transformation. The improved prognostic evaluation of patients affected with CLL entails genetic and cytogenetic markers, such as the mutation of tumor protein 53 (TP53), the deletion of the short arm of chromosome 17 (del17p) and the mutational status of immunoglobulin heavy chain variable region (IgHV). Further indicators of dismal course include the mutation of neurogenic locus notch homolog protein 1 (NOTCH1) and splicing factor 3b subunit 1 (SF3B1) and a complex karyotype [3,4]. 

Despite the improved management of CLL achieved over the past years [5], the persistence of CLL cells after therapy (i.e., minimal residual diseases, MRD) ultimately leads to disease relapse [6], which holds true also in the current era of new CLL-targeted therapies, including Bruton’s Tyrosine Kinase inhibitors (BTKi) (ibrutinib and acalabrutinib), B-cell lymphoma 2 (BCL2) inhibitors (venetoclax) and phosphoinositide 3-kinase inhibitors (idelalisib) [7,8,9,10,11,12,13,14,15]. This calls for further understanding of CLL biology and for the development of complementary therapeutic approaches. 

Despite being recognized as a low-mutational burden tumor entity [16], CLL appears to represent a suitable target for T cells and T cell-based immunotherapies, as documented by the graft vs. leukemia effect after allogenic bone marrow transplantation [17,18] and by cases of spontaneous remission after viral infection [19]. The immunogenicity of CLL and the possibility of harnessing the immune system to eliminate CLL cells require a deeper understanding. Tumor antigens are constituted of human leukocyte antigen (HLA) class I and HLA class II-presented peptides, which are recognized by CD8^+^ and CD4^+^ T cells, respectively [20]. We previously demonstrated that beyond neoepitopes also non-mutated tumor-associated antigens, which result from differential gene expression or protein processing in the tumor cells, could induce pathophysiologically relevant T cell activation in CLL [21]. Accordingly, the mass spectrometry-based analysis of the entirety of HLA-restricted peptides presented on tumor cells (i.e., immunopeptidome) enabled the identification of high frequent tumor-associated antigens for various tumor entities [16,21,22,23,24,25,26]. In line with these findings, we initiated two clinical trials (NCT02802943 and NCT04688385) with multi-peptide vaccines for CLL patients undergoing different therapeutic strategies, in order to elicit a specific T cell activation against CLL cells [27,28]. So far, mainly the pathophysiological role of antigen-specific T cell responses on disease outcome was investigated [21,29], whereas less is known about the effect of the diversity of the immunopeptidome, i.e., the number of different HLA-restricted peptides presented on malignant cells and identified by mass spectrometry. Recently, we could demonstrate that the HLA class II-restricted presentation of high numbers of different tumor antigen-derived peptides is associated with improved clinical outcome in ovarian cancer patients [30]. In this study, we now investigated the impact of the diversity of the immunopeptidome on clinical features and disease outcome of 57 CLL patients. Thereby, we could show that higher numbers of total and CLL-associated HLA class I-restricted peptides associate with a more favorable disease course.

## 2. Materials and Methods

### 2.1. Immunopeptidome Data

Immunopeptidome data were retrieved from a previous publication [16]. For immunopeptidome analysis peripheral blood mononuclear cells (PBMCs) from CLL patients were isolated. The immunopeptidome dataset comprised the total number of identified HLA class I- and HLA class II-presented peptides per patient, as well as the number of CLL-exclusive HLA class I- and HLA class II-restricted peptides per patient. CLL-exclusive peptides were defined as peptides presented exclusively in the immunopeptidome of CLL patient samples and never on any benign tissue sample, as demonstrated by comparative immunopeptidome profiling with more than 300 different benign tissues. CLL-exclusive peptides included only non-mutated CLL-associated antigens, as no neoepitopes were identified in the immunopeptidome analysis. Since two different mass spectrometers (LTQ Orbitrap XL and Orbitrap Fusion Lumos) were utilized for the immunopeptidome analysis, the numbers of identified peptides acquired with the LTQ Orbitrap XL mass spectrometer were normalized to those obtained with the Orbitrap Fusion Lumos mass spectrometer based on the results of 20 samples measured on both mass spectrometers. The number of normalized peptides per patients and, therefore, the individual peptide diversity ranged from 1186 to 9530 peptides for HLA class I peptides, from 935 to 12,282 for HLA class II peptides, from 155 to 3726 for CLL-exclusive HLA class I peptides and from 28 to 5780 for CLL-exclusive HLA class II peptides. 

### 2.2. Clinical Data

Clinical and survival data of patients affected with CLL, whose malignant cells had been previously analyzed by immunopeptidome analysis [16], were collected at the University Hospital Tübingen with a follow-up phase of up to 10 years after the primary diagnosis. Diagnosis was based on the iwCLL guideline [3]. Classification of CLL cases was performed according to the Binet staging system [3,4,31]. Cytogenetic analyses were performed with standardized methods at the medical care center (MVZ) Dortmund laboratory. Informed consent was obtained in accordance with the Declaration of Helsinki protocol. The study was approved by and performed according to the guidelines of the local ethics committees (373/2011B02, 454/2016B02, 406/2019BO2). In order to record the numbers of disease’s progression or relapses, any new disease’s presentation requiring CLL-specific treatment according to iwCLL guidelines [3] was documented for each patient. Patients were classified into two groups according to the number of relapses, namely patients with ≤1 relapse (group 1), i.e., presenting one or no progression/relapse after the first diagnosis, regardless of therapy indication at the moment of the first diagnosis, and patients with ≥2 relapses (group 2), i.e., presenting two or more relapses after the first diagnosis, regardless of therapy indication at the moment of primary diagnosis.

### 2.3. Software and Statistical Analysis

Data are displayed as box plots including median, 25th and 75th quartiles and min/max whiskers. Continuous data were tested for distribution and individual groups were tested by use of Kruskal–Wallis or Mann–Whitney U-test. If applicable correction for multiple comparison was done. Functional classification was performed of the source proteins of HLA class I-presented peptides by PANTHER 17.0 [32,33]. Progression-free (PFS) and overall survival (OS) were calculated by the Kaplan–Meier method. Log-rank test was performed to assess the difference of survival between groups. For survival analysis, patients were classified in a “high” or “low” group according to a calculated cut-off (≤“low” group, >“high” group) for the number of HLA-presented peptides (Appendix A). For predictive cut-off value estimation, we performed receiver-operating characteristics (ROC) analysis for each HLA class with respect to PFS and OS and value of highest Youden index was used as cut-off. Graphs were plotted using GraphPad Prism v.9.1.2. Statistical analyses were conducted using JMP Pro (SAS Institute, v.14.2) software. *p* values ≤ 0.05 were considered statistically significant.

## 3. Results

### 3.1. Characteristics of Patient Cohort

We analyzed the clinical, genetic and survival data of 57 CLL patients (Table 1 and Appendix A) in relation to HLA-restricted antigen presentation within the HLA class I and HLA class II immunopeptidome [16]. Patients’ median age at diagnosis was 61 years (range 38–90 years). In total, 70% of the patients were male. The frequency of patients presenting stage A, B and C according to the Binet staging system was 56%, 30% and 14%, respectively. Additionally, 58% of patients presented unmutated IgHV, while 22% of the subjects included in the study presented either a TP53 mutation or a del17p. Patients were followed up for up to 10 years after the first diagnosis and accompanying diseases were further defined according to the presence of autoimmune phenomena (14%), secondary tumors (25%) and hypogammaglobulinemia (26%), as documented throughout the entire observational period of time. The median PFS and OS were 45 and 104 months, respectively (Table 1). Overall, 81% of the patients were therapy-naïve at the time of the sample collection and the mean frequency of CLL cells in peripheral blood was 84%. 

### 3.2. Association of HLA-Restricted Antigen Presentation with Clinical Characteristics

The association of clinical characteristics of CLL patients with peptide presentation was investigated using mass spectrometry-derived immunopeptidome data of total as well as CLL-exclusive HLA class I- and HLA class II-restricted peptides. The presentation of peptides did not differ among patients with different Binet stages assessed at the primary diagnosis (Figure 1A). Furthermore, we analyzed the association between genetic and cytogenetic markers of unfavorable prognosis with the immunopeptidome diversity to investigate if immunopeptidome-driven tumor surveillance is hampered in high-risk patients. However, no differences in total or CLL-exclusive HLA peptide presentation were observed for the mutational status of IgHV (Figure 1B) and the presence of a TP53 mutation or del17p (Figure 1C). 

The clinical phenotype of the patients was characterized in more detail and the presence of autoimmune phenomena (including autoimmune cytopenia, psoriasis, vasculitis, and sarcoidosis), secondary tumors (including skin, bladder, colon, pancreatic, breast, and prostate cancer) and hypogammaglobulinemia was analyzed in relation to the immunopeptidome data. No difference in terms of total and CLL-exclusive number of peptides presented on CLL cells was observed among patients with or without any of these clinical features (Appendix A). 

### 3.3. A More Diverse Immunopeptidome Associates with Fewer CLL Relapses

We investigated the association of immunopeptidome diversity and the disease’s course in terms of CLL relapses. Patients were classified into two groups, according to the number of relapses (≤1 or ≥2 relapses). Patients with CLL cells presenting a more diverse immunopeptidome in terms of higher numbers of total and CLL-exclusive HLA class I-restricted peptides, experienced no more than one disease progression/recurrence compared to patients with a less diverse CLL-derived immunopeptidome who experienced multiple relapses (*p* = 0.015 and *p* = 0.002, respectively, Figure 2A). Of note, when considering only the smaller subgroup of therapy-naïve patients, similar results were observed for CLL-exclusive HLA class I peptides (*p* = 0.042) (Appendix A). Functional classification of the source proteins of the HLA-presented peptides revealed that the peptides originated from proteins that participate in similar pathways in both patient groups, including for example Wnt signaling, thus supporting that not only the peptide origin, but also the number of different antigens impact immunosurveillance. In line with HLA class I data, the presentation of higher numbers of CLL-exclusive HLA class II peptides is associated with a lower number of relapses (*p* = 0.047, Figure 2B). For total HLA class II-restricted peptides, a similar trend was observed without reaching statistical significance (Figure 2B and Appendix A).

### 3.4. A More Diverse HLA Class I Immunopeptidome Associates with Improved PFS

Subsequently, we investigated the impact of immunopeptidome diversity on PFS and OS. For total and CLL-exclusive HLA class I-restricted peptide presentation, we observed a significantly improved PFS in patients with CLL cells presenting higher numbers of these peptides (*p* = 0.027 and *p* = 0.022, respectively, Figure 3A,B). Even though deaths were only observed in the group of patients with a less diverse immunopeptidome (7 cases), no significant difference in the OS between the two groups was seen in the follow-up period of 10 years. For HLA class II-restricted peptide presentation, PFS and OS did not differ between patient groups with a low or high immunopeptidome diversity regardless of total or CLL-exclusive HLA peptides (Figure 4).

## 4. Discussion

Immunosurveillance of malignant cells by T cell-based recognition of HLA-presented peptides is a central mechanism of physiological tumor control [34]. Highly immunogenic malignancies, i.e., tumor entities characterized by neoepitopes arising from tumor-specific mutations, can elicit an efficient immune reaction, which can ultimately limit tumor growth and spread [35,36]. This unveiled intense efforts in order to enhance tumor-specific immune reactions and the field of immunotherapy rapidly expanded in the past years with breakthrough therapeutic innovations that include immune checkpoint inhibitors, bispecific antibodies, vaccinations and adoptive T cell transfer [37,38,39,40,41]. These novel therapies relay on T cell-mediated cytotoxicity based on T cell recognition of HLA-presented tumor antigens. Recently, several groups including ours uncovered tumor-associated non-mutated self-peptides as additional targets for T cell-based immunosurveillance [16,21,42,43,44], thus suggesting a possible role for T cell-based immunotherapies also in malignancies with low mutational burden, e.g., CLL. Accordingly, the mass spectrometry-based characterization of the entirety of naturally presented peptides, the so-called immunopeptidome, is being increasingly implemented in order to define novel tumor-associated antigens and expand therapeutic targets [16,21,22,23,24,25,26,45]. In previous publications [16,21], we identified CLL-associated HLA class I- and HLA class II-presented tumor antigens and showed that non-mutated CLL-associated peptides elicited a pathophysiologically relevant CLL-specific T cell activation [21]. This allowed us to develop patient-specific peptide vaccines against CLL cells, which served as a basis for two ongoing clinical trials (NCT02802943 and NCT04688385) [27,28]. The importance of CLL-specific T cell-based immune recognition of mass spectrometry-defined tumor antigens was also demonstrated showing a survival benefit as well as a lower incidence of skin cancer in patients exhibiting a T cell-based immune response to those CLL-associated antigens [21,29]. However, even if we proved the influence of a more diversified CLL-specific T cell responses on the disease’s course, the impact of the CLL-derived immunpeptidome and, in particular, the diversity of the HLA-presented peptides on patients’ outcome has not been investigated, yet. 

Here, we reported on the association of HLA class I and HLA class II immunopeptidome diversity with clinical and genetic characteristics as well as clinical outcome of CLL patients.

Analyzing established prognostic markers such as Binet classification at diagnosis, no significant difference was observed for the diversity of HLA-presented peptides. This held true also for molecular risk parameters, including IgHV, del17p and TP53 mutational status [3], which does not support the hypothesis of a more efficient tumor escape driven by a less diverse immunopeptidome in patients carrying a high genetical risk. 

Furthermore, despite our previous observations on a lower incidence of pre-malignant and malignant skin lesions in patients presenting CLL-specific T cells [29], we found that the diversity of the immunopeptidome does not differ between subjects with and without secondary tumors. In line, for autoimmune phenomena and hypogammaglobulinemia no difference was observed. Altogether, these results point towards a negligible effect of the immunopeptidome on secondary tumors and autoimmune phenomena frequently associated with CLL [3,29].

In terms of disease outcome, we showed that patients with an indolent disease course (i.e., number of relapses ≤ 1) presented higher numbers of total and CLL-exclusive HLA class I- as well as CLL-exclusive HLA class II-restricted peptides on their malignant cells compared to patients who present an aggressive disease course. Despite the previously reported stability of the immunopeptidome in CLL patients undergoing therapy, it could be hypothesized that the diversity of the immunopeptidome in CLL decreases in the disease course, due to its clonal evolution [46] and the emergency of predominant low-antigenic clones at any relapse, which should be investigated in future multi-dimensional studies integrating genomics-based approaches. Of note, the source proteins of HLA-presented peptides mapped in similar pathways in both patient groups, highlighting the importance of the number of different HLA-presented antigens for immunosurveillance. However, the addition of proteomics analysis in further validation studies might be helpful. 

In line with the observed effect on relapse numbers, we identified a prolonged PFS in CLL patients with a high number of different HLA class I-presented peptides (total and CLL-exclusive) in comparison to patients with a less diverse CLL immunopeptidome. Despite the observed longer PFS of patients with a high number of HLA class I peptides, the HLA class I immunopeptdome showed no impact on patients’ OS after a follow-up of 10 years. To this regard, it is noteworthy that CLL is a chronic disease with OS ranging from 2 to 20 years [3,4]. Therefore, a longer follow-up period might detect the impact of the immunopeptidome on OS calling for further studies. Moreover, with routine application of novel agents such as ibrutinib and venetoclax, the historic long-term OS of CLL patients might be significantly improved [13]. 

In summary, the observations of this study together with previous functional characterization of CLL-associated peptides [16,21] seem to suggest an efficient CD8^+^ T cell-based immunosurveillance of CLL cells. In contrast, presentation of HLA class II-restricted peptides and, thus, CD4^+^ T cell-mediated immunosurveillance, which we recently found to associate with improved outcome in ovarian cancer [30], appears to play a subsidiary role, as an indolent disease course could not be confirmed by prolonged PFS. However, the preliminary observation that subjects with higher numbers of CLL-exclusive HLA class II-presented peptides experienced fewer relapses is a first hint for the importance of CD4^+^ T cell responses and warrants further analysis in a larger cohort of CLL patients. Notably, an impaired immunogenicity, driven, e.g., by either HLA downmodulation or homozygosity of HLA alleles, is a well-known mechanism of tumor escape that leads to a poorer response to check-point inhibitors and shorter survival in B-cell lymphoma and solid tumors [47,48]. To this regard, we previously showed how CLL cells do not downmodulate HLA class I nor HLA class II molecules compared to normal B cells [20], which sustains the concept of a CLL-intrinsic diversified immunoppetidome that results to be associated with patients’ outcome, consistently with what was described in other tumor entities [48].

These results corroborate our previous findings in CLL patients [16,21], questioning the notion of CLL as a poor immunogenic malignancy, thus supporting current efforts to develop T cell-based immunotherapies for CLL patients. Indeed, despite the improved OS obtained with novel CLL treatments, response to therapy is usually limited to a partial remission or complete remission with MRD positivity, which eventually leads to disease relapse [12,13]. Moreover, quality of life is constrained by permanent treatment with BTKis and limited by the onset of side effects. Therefore, an accompanying application of T cell-based treatments in combination with established CLL treatments should be further investigated as a promising perspective for patients with CLL.

## 5. Conclusions

In conclusion, our study demonstrates the association between the diversity of total and CLL-exclusive HLA class I-restricted peptides on CLL course and suggests a role of naturally presented HLA class I peptides for the immunosurveillance of CLL. This supports the further investigation of T cell-based therapies as complementary approach to the standard of care treatments, in order to improve patients’ quality of life and survival.

## Figures and Tables

**Figure 1 cancers-14-04659-f001:**
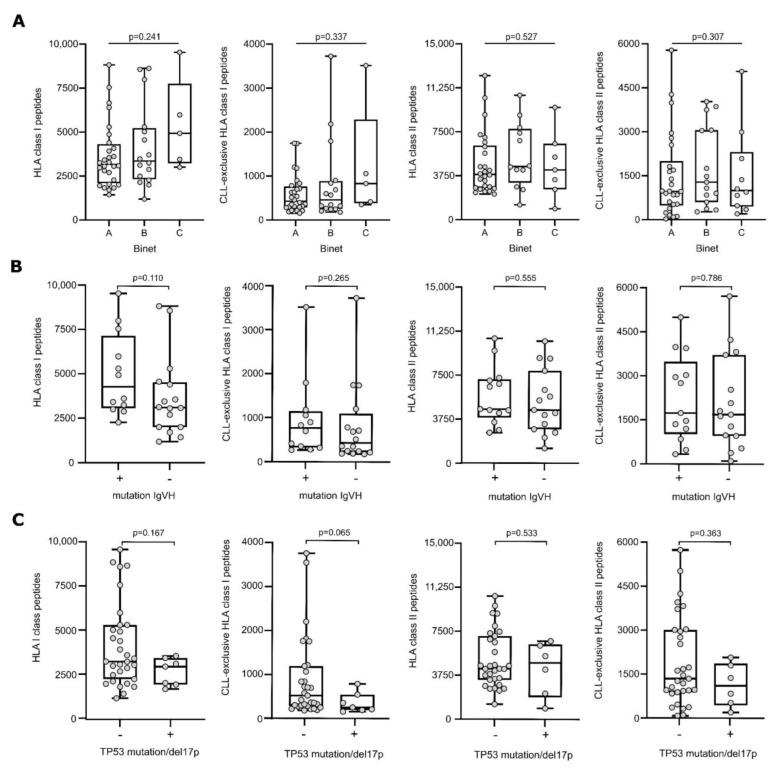
HLA presentation of antigenic peptides according to Binet stage and genetic background. Number of total HLA class I- (left panel), CLL-exclusive HLA class I- (mid-left panel), HLA class II-(mid-right panel), and CLL-exclusive HLA class II-presented (right panel) peptides according to the Binet classification (**A**), the mutational status of IgHV (**B**) and TP53 mutation/del17p (**C**). Dots represent data from individual patients. Boxes represent median and 25th to 75th percentiles with Min/Max whiskers. Brackets mark significant *p*-values between two categories (Kruskal–Wallis test with Dunn correction), continuous lines indicate non-significant *p*-values between all categories (Kruskal–Wallis test). Abbreviation: IgHV, immunoglobulin heavy chain variable region; TP53, tumor protein 53; del17p, deletion of the short arm of chromosome 17, *p*, *p*-value.

**Figure 2 cancers-14-04659-f002:**
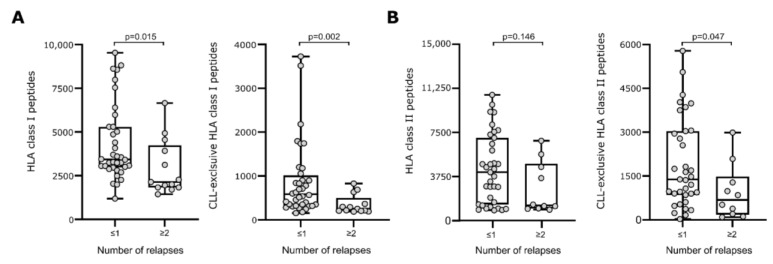
HLA presentation of antigenic peptides according to disease’s course in terms of observed relapses. Numbers of total (left panel) and CLL-exclusive (right panel) HLA class I- (**A**) and HLA class II-restricted (**B**) peptides according to the number of observed relapses. Dots represent data from individual patients. Boxes represent median and 25th to 75th percentiles. Brackets indicate *p*-values (Mann–Whitney U-test). Abbreviation: *p*, *p*-value.

**Figure 3 cancers-14-04659-f003:**
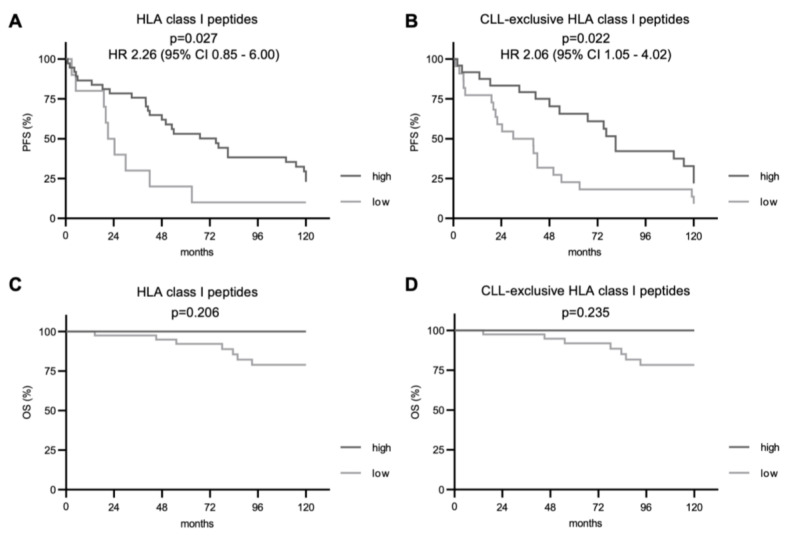
Impact of total and CLL-exclusive HLA class I peptide presentation on clinical outcome. Impact of total and CLL-exclusive HLA class I peptide presentation on PFS (**A**,**B**) and OS (**C**,**D**). Kaplan–Meier analysis, log-rank test. Abbreviations: HR, hazard ratio; CI, confidence interval; *p*, *p*-value.

**Figure 4 cancers-14-04659-f004:**
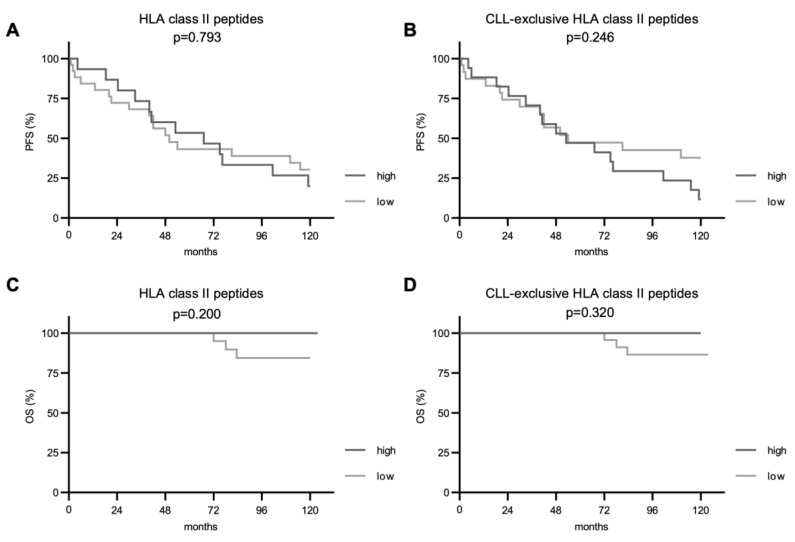
Impact of total and CLL-exclusive HLA class II peptide presentation on clinical outcome. Impact of total and CLL-exclusive HLA class II peptide presentation on PFS (**A**,**B**) and OS (**C**,**D**). Kaplan–Meier analysis, log-rank test. Abbreviations: *p*, *p*-value.

**Table 1 cancers-14-04659-t001:** Characteristics of CLL patient cohort.

	CLL Patients (n = 57)
Sex (n (%)) Male	40 (70)
Female	17 (30)
Age (years)	
Median Range	61 38–90
Binet (n (%)) A	32 (56)
B	17 (30)
C	8 (14)
Relapses (n (%))	
≤1	42 (74)
≥2	15 (26)
IgHV (n (%))	
Unmutated	18 (58)
Mutated	13 (42)
Unknown	26
TP53 mutation/del17p (n (%)) Negative	35 (78)
Positive	10 (22)
Unknown	12
Accompanying diseases Autoimmune phenomena Secondary tumors Hypogammaglobulinemia	8 (14) 14 (25) 15 (26)
PFS (months) Median	45
OS (months)	
Median	104

Abbreviations: n, number of patients; IgHV, immunoglobulin heavy chain variable region; TP53, tumor protein 53; del17p, deletion of the short arm of chromosome 17; PFS, progression-free survival; OS, overall survival.

## Data Availability

The data presented in this study are available in the article and supplementary material while further details can be obtained on request from the corresponding author.

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
