# Peer review of "Immunopeptidome Diversity in Chronic Lymphocytic Leukemia Identifies Patients with Favorable Disease Outcome"

_cancers, 2022, doi:10.3390/cancers14194659_

Round 1
Reviewer 1 Report
Marconato et al. presents the analysis of immunopeptidome datasets performed in a diverse set of human CLL samples. The authors identified a correlation between increased diversity in total and CLL-associated HLA class I peptides with better clinical outcome (fewer number of relapses) and overall survival. HLA class II antigen presentation was also studied, where only CCL-exclusive peptide diversity was linked with less relapses.
The manuscript provides a well-written and cited introduction and explains with sufficient detail the followed methodology. The strongest point of this study is the use of a substantial number of human CCL samples, in which immunopeptidomics was performed. This represents a solid and useful set of clinical data of substantial medical and scientifical value. Nevertheless, such dataset was published in a recent paper (Nelde et al, Front. Immunol., 2021), limiting its novelty contribution in this paper. In addition, the conclusion drawn by the authors -that T cell-based therapy application represents a promising therapy option- is not directly supported by the data, since the observed correlation is not causally demonstrated experimentally. Therefore, I would not recommend the publication of this paper unless the following major concerns are addressed:
- The study relies on the correlation of higher number of total and CCL-exclusive antigens presented by HLA class I and II in CCL samples with a better clinical outcome. However, this approach fails to address the physiological and functional relevance of this data. The demonstration of a causal effect of higher peptide presentation in overall survival would greatly improve the impact and relevance of the work.
- Further concerns are the non-specificity of bulk data, where the increase in the number of presented antigens cannot be attributed only to a certain type of cell, such as malignant or healthy cells, or to type of antigen presenting cell (DC, B cell, macrophage…). Most importantly, the CLL-specific antigen increase in diversity could be result of an increase in non-CLL cells, which would completely revoke the paper’s premise.
- The moment of the disease evolution in which the samples were taken and analyzed could be crucial for data interpretation. In this way, it would be reasonable to think that treatment and functioning of the immune system would end up selecting clones with less diversity, which will survive and cause relapse. Therefore, if samples from patients in early stages of the disease were analyzed together with patients with advanced disease progression, it is logical that those patients in advanced stages, and who therefore have presented more than one relapse, present less antigenic diversity (since the immune system and the treatment have eliminated those clones with greater antigenic diversity in the early stages of disease). A study comparing antigen diversity of individuals in their early stages of disease versus advanced stages might be insightful.
- There is a lack to define which peptides and pathways are differently upregulated in the low-relapse group of patients. Given the availability of data, a more exhaustive study of peptide signatures could be potentially revealing.
- The paper doesn’t report which peptides are considered CLL-exclusive antigens and doesn’t justify what reasoning was followed to define them as such.
Some minor points:
- The section 3.3 does not provide enough solid justification to affirm in its title that “diverse immunopeptidome prevents CLL relapse”. Stating an “association” would be in line with the data provided.
- A justification of why it was decided to look at the correlation of different mutational states with HLA presentation (Figure 1C) is desirable (line 129).
- A more extensive explanation of how high and low antigen diversity ranges were established (section 2.3) would be appreciated.
Overall, the manuscript provides a valuable dataset of human CLL immunopeptidomes that could potentially reveal crucial insights into antigen diversity and presentation in CLL. Nevertheless, the conclusions drawn from the observed data and correlations established are not supported by experimental causal demonstration, thus diminishing its physiological and translational relevance.
Reviewer 2 Report
Marconato et. al. in this article has analyzed publically available dataset to find potential biomarker and its importance Chronic lymphocytic leukemia (CLL). These potential biomarker studies definitely show some indications but a downstream validation or confirming evidence is absolutely needed to have confidence with the claims.
Overall, this study on its own does not provide any significant information/knowledge to the broader scientific community. Therefore, I think this is not suitable for publication
Reviewer 3 Report
M. Marconato et al. describe an analysis of the relationships between immunopeptidomes of 57 CLL patients, molecular and clinical features, and disease outcome measured as number of relapses, progression-free survival and overall survival. Among the parameters tested, results identified a relationship between high numbers of HLA-I-presented peptides (total and CLL-exclusive) and favorable disease outcome (<1 relapse and longer progression-free survival). High numbers of CLL-exclusive HLA-II peptides characterized patients with <1 relapse. The Authors propose that these findings confirm the existence of robust T-cell-mediated immunosurveillance of CLL cells and support a rationale for T-cell-based immunotherapies.
This concise, well-written manuscript presents important findings that are pertinent to the special topic.
A few points need to be addressed.
The graphs in Figures 1 and 2 show many cases that lie outside the boxes. It would be useful to add a supplementary table that summarizes each patient’s molecular/clinical characteristics together with the individual HLA I and II immunopeptidome values.
On lines 125-127, the Authors should indicate the cut-off values for classifying ‘high’ and ‘low’ groups.
Line 179 refers to ‘Figures S1’. This Reviewer could not find the figure.
Did the Authors compare expression of HLA-I and HLA-II molecules in these samples to see if this correlates with the immunopeptidome differences and prognosis? In this regard, the Authors could point out that a previous analysis indicated that CLL cells do not downregulate HLA expression when compared to normal B-cells (ref. 17 in the manuscript).
On line 274 surveillance is spelled wrong.
On line 303, what does ‘visualization’ mean- preparation of table and figures?
Reference 27 (a 2018 review on CLL) should be moved to the first paragraph of the Introduction.
Lines 57- 58 address the problem of minimal residual disease. Citation of ref. 4 on line 58 seems inappropriate, as this paper investigates alemtuzumab; ref. 4 could be grouped together with refs. 5-11. The Authors might want to consider reworking the text and citing the review by Albiol N et al., Current Opinion in Oncology 2021;33:670 that describes CLL therapies, including their impact on MRD.
Round 2
Reviewer 1 Report
Authors have answered all my questions and I feel the article meets all the publication criteria
Reviewer 2 Report
I do not see any improvement in the revised manuscript and the authors did not address any of the concerns raised.
I do not think this article is ready for publication